# Psychometric properties of the culturally adapted 10-item Hopkins Symptom Checklist (HSCL-10-SW) anxiety subscale for southwestern Madagascar[‡]

Hervet J. Randriamady[1,2,3] (ORCID), Manasi Sharma[4,5,6], Rocky E. Stroud II[4], Aroniaina M. Falinirina[7], Romario[7], Madeleine Rasoanirina[7], Nadège V. Volasoa[8], Frédéric Déclerque[7], Marc Y. Solofoarimanana[7], Jean C. Mahefa[7], Hanitra O. Randriatsara[9], Karestan C. Koenen[4,10] and Christopher D. Golden[2,3,11,12]

[1]Harvard Kenneth C. Griffin Graduate School of Arts and Sciences, Cambridge, MA, United States; [2]Department of Nutrition, Harvard TH Chan School of Public Health, Boston, MA, United States; [3]Madagascar Health and Environmental Research (MAHERY), Maroantsetra, Madagascar; [4]Department of Epidemiology,Harvard TH Chan School of Public Health, Boston, MA, United States; [5]RAHAT Charitable and Medical Research Trust, New Delhi, India; [6]Heidelberg Institute of Global Health (HIGH),Faculty of Medicine and University Hospital, Heidelberg University, Heidelberg, Germany; [7]Institut Halieutique et des Sciences Marines (IHSM), University of Toliara, Toliara, Madagascar; [8]Service de District de la Santé Publique, Ministère de la Santé Publique, Toliara, Madagascar; [9]Centre Hospitalier Universitaire des Soins et de Santé Publique Analakely (CHUSSPA), Service de la Formation et la Recherche (SFR); [10]Department of Social Behavioral Sciences, Harvard TH Chan School of Public Health, Boston, MA, United States; [11]Department of Environmental Health, Harvard TH Chan School of Public Health, Boston, MA, United States and [12]Department of Global Health and Population, Harvard TH Chan School of Public Health, Boston, MA, United States

## Research Article

**Keywords:**
anxiety disorders; depression; psychometric properties; Vezo; Masikoro; Madagascar; cultural adaptation

**Corresponding author:**
Hervet J. Randriamady;
Email: hrandriamady@g.harvard.edu

[‡]This article has been updated since original publication. A notice detailing the change has been published.

## Abstract

Mental health conditions, including anxiety disorders, are a major cause of morbidity across Sub-Saharan Africa. There are scarce mental health resources and providers in Madagascar, which substantiates a need for clear and accessible assessment tools for assessing mental health conditions. Yet, before this study, there were no validated scales to measure anxiety disorder symptoms in Madagascar. We assessed the psychometric properties of the culturally adapted 10-item Hopkins Symptom Checklist (HSCL-10-SW) anxiety subscale in the Bay of Ranobe region, in southwestern Madagascar. The study participants were part of the ongoing HIARA cohort study. The HSCL-10-SW includes the original HSCL-10 anxiety subscale in addition to three culturally relevant items that were derived through qualitative research: *irritability*, *lost in thoughts/overthinking* and *forgetfulness*. We administered the HSCL-10-SW to 809 participants (41.2% males) aged 16 years (mean age 36.9) and above in October 2023. Our exploratory factor analysis supported a two-factor structure: Fear Anxiety and Cognitive-Somatic Anxiety. We found discriminant validity between Fear anxiety and Depression factors. Although the HSCL-10-SW demonstrated acceptable psychometric validity, we suggest that additional qualitative studies should be conducted to explore the local conceptualization of anxiety disorders in southwestern Madagascar.

## Impact statement

Madagascar lacks mental health care specialists, with only 24 psychiatrists for ~30 million people. Specifically, the southwestern region of Madagascar has only one psychiatrist for roughly two million people, indicating a tremendous need for mental health support. To acknowledge mental health needs in this region, we constructed the HSCL-10-SW, an adapted version of the Hopkins Symptom Checklist, as the first validated scale to measure symptoms of anxiety disorders in Madagascar. The HSCL-10-SW will help nonmental health specialists assess the needs and treatment gaps in southwestern Madagascar. Evaluating the degree of mental health need in this region is the first step to proactively develop financing, policies and tools to support people suffering from mental health conditions.

## Introduction

Anxiety disorders and depressive disorders, affecting roughly 359 and 332 million people, respectively, represent the two most common mental disorders globally (World Health Organization, 2025a, 2025b). Approximately 4.4% of the world population lived with an anxiety disorder in 2021, and only one out of four people with an anxiety disorder received treatment

(WHO, 2025b). During the COVID-19 pandemic between 2020 and 2021, nearly half of the population in more than 18 African countries experienced anxiety (Bello et al., 2022). Despite the high prevalence and treatment gap, research on the measurement of anxiety and depression in Sub-Saharan Africa remains limited.

The 25-item Hopkins Symptom Checklist (HSCL-25) is a commonly used scale that screens for general distress, depressive and anxiety symptoms (Derogatis et al., 1974). The 10-item Hopkins Symptom Checklist (HSCL-10) is a subscale of the HSCL-25 used to screen for anxiety disorders. Previous studies in Sub-Saharan Africa (Tanzania, Uganda, South Africa and Mali) have examined the psychometric properties of the HSCL-10 anxiety subscale, but only as part of the whole HSCL-25 scale (Kaaya et al., 2002; Lee et al., 2008; Ashaba et al., 2018; Lasater et al., 2022; Wolfaardt et al., 2024). Other studies have focused on culturally adapting the HSCL-25 by adding or removing items beyond the original scale in Africa to accommodate the local context. However, these adaptations have mainly focused on the depression subscale, especially among people living with HIV, pregnant women attending antenatal care or those affected by conflict in Sub-Saharan Africa (Bolton, 2001; Kaaya et al., 2008; Lasater et al., 2022). Scale measuring fear has been validated and developed in many cultural contexts (Shuja et al., 2022). Importantly, no studies have attempted to culturally adapt the HSCL-10 anxiety subscale itself by incorporating locally relevant symptoms in Sub-Saharan Africa.

Only three descriptive epidemiologic studies have been conducted on anxiety disorders in Madagascar. One study assessed the prevalence of clinically diagnosed anxiety disorders using the diagnostic criteria of the Diagnostic and Statistical Manual of Mental Disorders, Fourth Edition (DSM-IV) (American Psychiatric Association, 1998). They used a nonrandom sample of 1,576 inpatients and outpatients over 16 years in public and private hospitals in the capital city, Antananarivo, between November 2016 and June 2017 (Bakohariliva et al., 2020). The authors found that the prevalence of clinical anxiety disorders was 2% ($n = 27$ inpatients and $n = 5$ outpatients) for the whole sample. Among those diagnosed with anxiety disorders, 47% were jointly diagnosed with depression. Of the 2% diagnosed with anxiety disorders, PTSD was the most prevalent anxiety disorder, comprising 28% of the total, followed by panic disorder (22%) and generalized anxiety disorder (GAD; 19%). Roughly 16% of the patients diagnosed with anxiety disorders had anxiety disorders in the past, and 63% had experienced stressful life events (Bakohariliva et al., 2020). Another cross-sectional study conducted between July 2020 and October 2021 evaluated the prevalence of depressive and anxiety disorders among nonrandomly sampled inpatient women ($N = 36$) with gynecological and breast cancers in a public hospital in Toliara, in southwestern Madagascar (Randriamalala et al., 2024). The authors found prevalence rates of 39.40% for anxiety disorders and 45.46% for depressive disorders using the Hospital Anxiety and Depression Scale (HADS). A recent mixed-method study examined the association between climate change and the mental health of adolescents in southern Madagascar (Hadfield et al., 2025). The participants (49 females and 34 males) were recruited through convenience sampling. All participants lived in poverty and, on average, experienced a self-reported severe food insecurity level. Using the GAD-7 scale, the authors found that ~76.3% and 86.8% of adolescents met the threshold for probable GAD based on cut-off scores of 8 and 9, respectively (Hadfield et al., 2025). However, neither the GAD-7 nor the HADS has been validated in Madagascar, and there has yet to be a study evaluating the prevalence of anxiety disorders in a randomly sampled population in a nonclinical community setting.

Before this study, there were no culturally validated and adapted scales specifically designed to screen for anxiety disorders in Madagascar. The only existing validated and culturally adapted mental health scale in the country is the 8-item Patient Health Questionnaire (PHQ-8), which screens for depression and measures depressive symptoms (Randriamady et al., 2025). However, this scale was only validated in the southwestern region of Madagascar as part of an ongoing population-based study (Golden et al. 2024a). The need for cultural adaptation is crucial because scales developed in Western countries are based on the expression of emotions and feelings of their population and often fail to capture symptoms unique to other non-Western cultures (Bass et al., 2007). As a result, salient culturally specific symptoms might not be captured, and may lead to biased estimates of anxiety disorders. To address this gap, our study aims to adapt and culturally validate the original HSCL-10 anxiety subscale from the HSCL-25 by supplementing it with salient local symptoms identified through a mixed-methods study (Randriamady et al., 2025).

## Methodology

### *Study participants*

We enrolled 809 study participants aged 16 years and older from 14 communities as part of the Health Impacts of Artificial Reef Advancement (HIARA) cohort study in the Bay of Ranobe in southwestern Madagascar (Golden et al. 2024a; Randriamady et al., 2025). Vezo and Masikoro are the predominant ethnicities in the area. Vezo's main livelihood is tied to traditional small-scale fishing-related activities, whereas Masikoro are more pastoralists. In the HIARA cohort study, the 14 communities included 12 coastal communities ($n = 30$ households each) and two inland communities ($n = 45$ households each) for a total of 450 households. Detailed information on the study design can be found at Golden et al. (2024a) and Randriamady et al. (2025). Mental health surveys were administered every 3 months beginning in October 2023–January 2026. All participants verbally consented to participating in the study following Harvard TH Chan School of Public Health's IRB and locally approved protocols. At the beginning of the research, all adult participants signed a consent document following a reading and explanation of the consent document at the time of research enrollment. Following the first visit, all household members provided verbal consent to participate in the research (Golden et al., 2024a) (Table 1).

### *Cultural adaptation and translation of the original HSCL-10 anxiety subscale*

The cultural adaptation of the scale was done in three steps. First, we assessed the factor structure of the HSCL-10 anxiety subscale, which included the additional items *irritability*, *lost in thoughts/overthinking* and *forgetfulness* (HSCL-10-SW), using an exploratory factor analysis (EFA). Second, using the factor structure we obtained from the EFA, we conducted confirmatory factor analysis (CFA) to assess the factor structure. And finally, we evaluated the convergent and discriminant validity of the HSCL-10-SW with the validated PHQ-8 administered to the same participants.

### *Addition of three additional items: "Irritability," "lost in thoughts or overthinking" and "forgetfulness"*

Using results from our mixed-method study that aimed to elicit local mental health syndromes in the Bay of Ranobe in southwestern

**Table 1.** Sample characteristics (*N* = 809)

|  | Overall |
|---|---|
|  | *N* = 809 |
| **Sex** | |
| Female | 476 (58.8%) |
| Male | 333 (41.2%) |
| **Age (years)** | |
| Median | 33 |
| Mean (SD) | 36.9 (16.1) |
| **Age group (years)** | |
| 16–29 | 333 (41.2%) |
| 30–44 | 251 (31.0%) |
| 45–59 | 141 (17.4%) |
| 60+ | 84 (10.4%) |
| **Marital status** | |
| Married | 553 (68.4%) |
| Single | 188 (23.2%) |
| Separated | 48 (5.9%) |
| Widowed | 20 (2.5%) |
| **Ethnicity** | |
| Vezo | 408 (50.4%) |
| Masikoro | 160 (19.8%) |
| Antandroy | 123 (15.2%) |
| Other/Mixed | 118 (14.6%) |

Madagascar in 2022 and 2023 (Randriamady et al., 2025), we added three items to the HSCL-10 anxiety subscale: *irritability*, *lost in thoughts/overthinking* and *forgetfulness*. During this research, we conducted focus group discussions, free listing interviews and cognitive interviews with key informants to elicit these local syndromes. We found that *irritability* was a common symptom in depressive, grief-like and general distress syndromes. *Forgetfulness* was only a symptom associated with the general distress-like syndrome, whereas *lost in thoughts/overthinking* was associated with both the depressive and general distress-like syndromes (Randriamady et al., 2025).

### Translation of the HSCL-10 anxiety subscale

First, HJR and NVV translated the original HSCL-10 anxiety subscale items into the Vezo and Masikoro dialects, as these are the most commonly spoken dialects in the Bay of Ranobe in southwestern Madagascar. The two dialects are closely related, with differences primarily observed in intonation and a few vocabulary words. However, the Antandroy ethnicity, part of the study, has its own unique dialect. The Antandroy dialect also belongs to the broader southwestern Malagasy dialect group and shares linguistic similarities with Vezo and Masikoro (Adelaar, 2013; Serva & Pasquini, 2020). The Antandroy communities included in this study migrated to the Bay of Ranobe area in the 1940s and communicate fluently in Vezo and Masikoro. Second, HJR and NVV conducted cognitive interviews with 23 key informants (14 female and 9 male primary mental health care providers) who were knowledgeable about mental health in the the region to assess the comprehension of the translated scale (Randriamady et al., 2025). Third, adjustments

were made based on the feedback and suggestions from the key informants, including slight semantic modifications. For instance, in item 4 ("Nervousness or shakiness inside"), only the term "nervousness" was retained, and in item 7 ("Feeling tense or keyed up"), the phrase "keyed up" was dropped. In addition, there was no direct equivalent of the word "nervousness" in the Vezo and Masikoro dialects or the Malagasy language. Thus, we used the two words *Taitaitsy* or *Tofotofotsy*, which literally mean "startled" or "shocked" and can encompass anxiety, worry and fear. The cognitive interviews with key informants also revealed that some translated items were interpreted differently from their intended meaning, necessitating further adjustment. For example, item 7 ("Feeling tense") was frequently understood as referring to bodily strength or seizures (often linked to epilepsy), and during back-translation, it was again translated as "vigorous body" by one translator. Thus, HJR and NVV tested multiple alternative phrasings until item 7 was no longer associated with strength or seizures. Similarly, in item 9 ("Spells of terror or panic"), the word "panic" was commonly interpreted as being "in a hurry" or "rushing" in both cognitive interviews with key informants and back-translation. Consequently, "panic" was dropped, and only "spells of terror" was retained.

### Measures

#### The 10-item Hopkins Symptom Checklist for southwestern Madagascar (HSCL-10-SW)

The HSCL-10-SW is a self-reported scale composed of the HSCL-10 anxiety subscale (Derogatis et al., 1974) from the 25-item Hopkins Symptom Checklist (HSCL-25), supplemented by three additional culturally relevant items: *irritability*, *lost in thoughts/overthinking* and *forgetfulness*. Specifically, the HSCL-10 anxiety subscale consisted of 10 items: item 1 ("Suddenly scared for no reason"), item 2 ("Feeling fearful"), item 3 ("Faintness, dizziness or weakness"), item 4 ("Nervousness"), item 5 ("Heart pounding or racing"), item 6 ("Trembling"), item 7 ("Feeling tense"), item 8 ("Headaches"), item 9 ("Spells of terror") and item 10 ("Feeling restless, can't sit still"). The three new items added to the HSCL-10 anxiety subscale are: item 11 ("Irritability"), item 12 ("Lost in thoughts/overthinking") and item 13 ("Forgetfulness"). We asked the number of days in the past week (7 days) the participants had experienced each of the 13 items: 0 days (1, "Not at all"), 1–3 days (2, "A little bit"), 4–5 days (3, "Quite a bit") and 6–7 days (4, "Extremely"). The total score ranges from 13 to 52.

#### The 8-item Patient Health Questionnaire (PHQ-8)

The 8-item Patient Health Questionnaire (PHQ-8) is a self-reported scale to screen for major depressive disorder (Kroenke et al., 2009). The one-factor PHQ-8 model has been validated using the same study participants from this cohort in southwestern Madagascar (Randriamady et al., 2025).

### Statistical analysis

#### Reliability of the HSCL-10-SW items

We computed the Cronbach's alpha and the ordinal alpha coefficients to assess the internal consistency of the HSCL-10-SW items. However, we primarily chose the ordinal alpha as a measure of internal consistency because the observed response variables were assumed to be ordinal for the remainder of the psychometric analysis (Zumbo et al., 2007; Gadermann et al., 2012).

## Factor analysis

We randomly split our sample into two groups: one for EFA ($n = 406$) and the other for Confirmatory Factor Analysis (CFA) ($n = 403$). We used the first half of the sample for EFA, a data-driven approach, to identify factors to be retained, and the second half for CFA, a theory-driven approach, to test hypotheses based on the factors identified in the EFA. We treated the data as ordinal. Thus, we used the Diagonal Weighted Least Squares (DWLS) estimator with robust standard errors and a mean-and-variance adjusted method for both EFA and first-order CFA. Consequently, polychoric correlations were used to estimate the association between the latent response variables. The latent response variables were preceded with an asterisk (e.g., HSCL-1*, HSCL-2* and HSCL-11-SW*) to distinguish them from the ordinal item variables (e.g., HSCL-1, HSCL-2 and HSCL-11-SW). We used delta parameterization by fixing the variance of common factors to 1 for CFA. We used the "lavaan" (Version 0.6–17), "semTools" (Version 0.5–6) and "polycor" (Version 0.8–1) packages in RStudio (Version 2024.09.0 + 375) for the analysis (Rosseel, 2012; RStudio Team, 2020; Jorgensen et al., 2022; Fox, 2025).

**Exploratory factor analysis.** To examine the adequacy of the sample ($n = 406$) before the EFA, we first evaluated the Kaiser–Meyer–Olkin (KMO) statistic (KMO = 0.80), which indicated an adequate sample size. Second, we conducted Bartlett's test of sphericity on Pearson's correlation matrix, which was statistically significant ($p < 0.001$), indicating that the matrix was not an identity matrix and that the data were suitable for EFA. Thus, based on the KMO statistic and Bartlett's test, we proceeded with the EFA. Third, we performed a scree plot analysis (Figure 1) to determine the number of factors to be retained before conducting the EFA. We retained factors with eigenvalues >1 for the EFA, based on inspection of the scree plot (Figure 1). Because the observed items were treated as ordinal indicators, we used polychoric correlations. Fourth, we conducted an EFA with oblique rotation to assess the dimensions of the HSCL-10-SW using a DWLS method with robust standard errors and a mean-and-variance approach. We retained standardized factor loadings >0.35.

**Confirmatory factor analysis.** We conducted a first-order CFA (Model 1.1) using half of the sample ($n = 403$) based on the EFA solutions we chose. Similar to the EFA, we used polychoric correlations and the DWLS method with robust standard errors, along with a mean-and-variance approach. To assess the model fit, we conducted a chi-square test. We also used other approximate model fit indices, such as the root mean square error of approximation (RMSEA), the standardized root mean square residual (SRMR), the Bentler Comparative Fit Index (CFI) and the Tucker–Lewis Index (TLI), because the chi-square test is sensitive to large sample sizes. CFI ≥ 0.95, TLI ≥ 0.95, RMSEA ≤0.08 and SRMR ≤0.08 were used as criteria for goodness-of-fit (Hu and Bentler, 1999; Kline, 2023). When model fit indices indicated poor fit, we respecified the model (Model 1.2) by inspecting the modification indices.

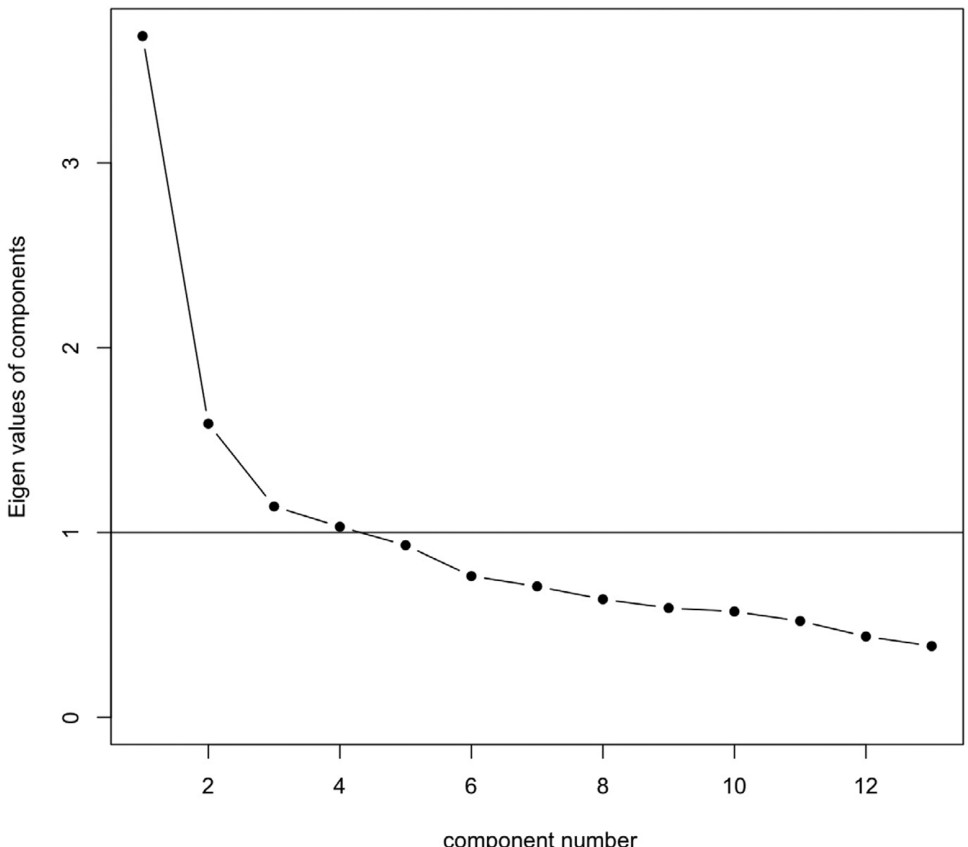

**Figure 1.** Scree plot ($n = 406$).

### Validity

We assessed the convergent and discriminant validity of the HSCL-10-SW using Cheung et al. (2023) and Rönkkö and Cho (2022). We used the full sample (*N* = 809) for the discriminant validity assessment.

Convergent validity. We evaluated the convergent validity of the HSCL-10-SW by following the recommendations of Cheung et al. (2023). That is, all standardized factor loadings should have a 90% upper limit confidence interval (ULCI) of >0.5, the average variance extracted (AVE) should have a 90% ULCI of >0.5 and the hierarchical omega coefficient should have a 90% ULCI of >0.7 (Cheung et al., 2023). However, we did not compute the hierarchical omega coefficient due to convergence issues. Instead, we used the omega coefficient subscale.

Discriminant validity. We assessed the discriminant validity of the HSCL-10-SW, using the full data (*N* = 809) to determine whether depression (PHQ-8) and anxiety disorders (HSCL-10-SW) were empirically distinct and did not measure the same underlying construct. More specifically, we hypothesized that the depression and anxiety disorder constructs were positively correlated, and the magnitude of the construct correlation would be <50%. We followed the operational definition and recommendations of Rönkkö and Cho (2022) and Cheung et al. (2023) to assess discriminant validity. That is, we first computed the polychoric correlation between the HSCL-10-SW and PHQ-8 latent response variables. Second, we conducted a first-order CFA for both the HSCL-10-SW and one-factor PHQ-8 (Model 2) as a whole scale and then assessed the overall model fit. However, instead of evaluating the 95% ULCI of the correlations between PHQ-8 and HSCL-10-SW models as recommended by Rönkkö and Cho (2022), we evaluated the 90% lower limit confidence interval (LLCI) as suggested by Cheung et al. (2023). The reason is that using the

95% ULCI of the correlation can reduce the power to detect discriminant validity. Therefore, a correlation with a 90% LLCI of <0.8 was the cut-off we used to assess discriminant validity (Rönkkö and Cho, 2022; Cheung et al., 2023).

## Results

### Sociodemographic characteristics and descriptive statistics of the HSCL-10-SW item

The majority of participants were female (58.8%), Vezo (50.4%) and married (68.4%). The median age of the participants was 33 years (Table 1). The endorsement of the HSCL-10-SW items ranged from 5 to 81% (Table 2). Item 12 ("Lost in thoughts or overthinking") was the most endorsed (81%), followed by item 11 ("Irritability"). These two most endorsed items were two items added to the original HSCL-10 anxiety subscale. Conversely, item 9 ("Spell of terror") was the least endorsed (5%).

### Reliability of the HSCL-10-SW items

The Cronbach's alpha (*α* = 0.78) and the ordinal alpha (*α* = 0.86) coefficients were above 0.7 (Table 2), which indicates that the HSCL-10-SW is a reliable scale.

### Exploratory factor analysis

The scree test analysis (Figure 1) ranged from one-factor to four-factor solutions for the EFA. However, after inspecting all solutions, we decided to retain the two-factor solution. We did not use additional statistical criteria for factor retention (e.g., parallel analysis) because such methods typically generate a similar range of factors (four to five), and factor selection was ultimately guided by

**Table 2.** HSCL-10-SW item mean scores and reliability coefficients (*N* = 809)

| Observed variables | Items | Mean | SD | Endorsement (%) | Skewness | Kurtosis |
|---|---|---|---|---|---|---|
| HSCL-1 | 1. Suddenly scared for no reason | 1.311 | 0.538 | 15 | 1.793 | 3.863 |
| HSCL-2 | 2. Feeling fearful | 1.561 | 0.817 | 40 | 1.517 | 1.742 |
| HSCL-3 | 3. Faintness, dizziness or weakness | 1.682 | 0.759 | 50 | 1.008 | 0.719 |
| HSCL-4 | 4. Nervousness | 1.577 | 0.661 | 37 | 1.023 | 1.147 |
| HSCL-5 | 5. Heart pounding or racing | 1.373 | 0.553 | 23 | 1.322 | 1.742 |
| HSCL-6 | 6. Trembling | 1.221 | 0.474 | 13 | 2.183 | 4.935 |
| HSCL-7 | 7. Feeling tense | 1.147 | 0.385 | 8 | 2.688 | 7.87 |
| HSCL-8 | 8. Headaches | 1.756 | 0.713 | 48 | 0.862 | 0.976 |
| HSCL-9 | 9. Spells of terror | 1.103 | 0.308 | 5 | 2.741 | 6.046 |
| HSCL-10 | 10. Feeling restless, cannot sit still | 1.438 | 0.596 | 38 | 1.152 | 1.005 |
| HSCL-SW-11 | 11. Irritability | 2.110 | 0.835 | 66 | 0.773 | 0.297 |
| HSCL-SW-12 | 12. Lost in thoughts/overthinking | 2.472 | 1.034 | 81 | 0.165 | −1.141 |
| HSCL-SW-13 | 13. Forgetfulness | 1.548 | 0.688 | 40 | 1.275 | 1.792 |
| | **Reliability coefficients** | | | | | |
| | Cronbach's alpha (*α*) | 0.78 | | | | |
| | Ordinal alpha (*α*) | 0.86 | | | | |

*Note:* 1 = endorsed ("A little," or "Quite a bit," or "Extremely"), and 0 = Not endorsed ("Not at all").

**Table 3.** DWLS with robust standard errors and mean-and-variance adjusted parameter estimates from EFA of the HSCL-10-SW with oblique rotation (*n* = 406)

| Latent response variables | Factor | Standardized loadings | Unique variances | Communalities |
|---|---|---|---|---|
| | **Fear Anxiety factor** | | | |
| HSCL-1* | 1. Suddenly scared for no reason | 0.473 | 0.725 | 0.275 |
| HSCL-2* | 2. Feeling fearful | 0.832 | 0.318 | 0.682 |
| HSCL-4* | 4. Nervousness | 0.681 | 0.440 | 0.560 |
| HSCL-5* | 5. Heart pounding or racing | 0.621 | 0.444 | 0.556 |
| HSCL-9* | 9. Spells of terror | 0.752 | 0.492 | 0.508 |
| | **Percent variance** | 20% | – | – |
| | **Cognitive-Somatic Anxiety factor** | | | |
| HSCL-3* | 3. Faintness, dizziness or weakness | 0.638 | 0.578 | 0.422 |
| HSCL-6* | 6. Trembling | 0.592 | 0.598 | 0.402 |
| HSCL-7* | 7. Feeling tense | 0.539 | 0.719 | 0.281 |
| HSCL-8* | 8. Headaches | 0.374 | 0.798 | 0.202 |
| HSCL-10* | 10. Feeling restless, cannot sit still | 0.403 | 0.664 | 0.336 |
| HSCL-11-SW* | 11. Irritability | 0.758 | 0.468 | 0.532 |
| HSCL-12-SW* | 12. Lost in thoughts/overthinking | 0.681 | 0.554 | 0.446 |
| HSCL-13-SW* | 13. Forgetfulness | 0.714 | 0.466 | 0.534 |
| | **Percent variance** | 24.20% | – | – |

theory, leading to two factors. The two factors (Table 3) explained 44.20% of the total variance. We labeled the first factor as Fear Anxiety and the second factor as Cognitive-Somatic Anxiety. The Fear Anxiety factor consisted of five items: HSCL-1* ("Suddenly scared for no reason"), HSCL-2* ("Feeling fearful"), HSCL-4* ("Nervousness"), HSCL-5* ("Heart pounding or racing") and HSCL-9* ("Spells of terror"), which accounted for 20% of variance. On the other hand, the Cognitive-Somatic Anxiety factor consisted of the eight items: HSCL-3* ("Faintness, dizziness or weakness"), HSCL-6* ("Trembling"), HSCL-7* ("Feeling tense"), HSCL-8* ("Headaches"), HSCL-10* ("Feeling restless, can't sit still"), HSCL-SW-11* ("Irritability"), HSCL-SW-12* ("Lost in thoughts/overthinking") and HSCL-SW-13* ("Forgetfulness"), which accounted for 24.20% of the variance (Table 3).

### Confirmatory factor analysis

Using the two-factor solution from the EFA (Fear Anxiety and Cognitive-Somatic Anxiety), we ran a first-order CFA (Model 1.1; Table 4), in which the Fear Anxiety and Cognitive-Somatic Anxiety factors were allowed to covary. Model 1.1 displayed an acceptable model fit (RMSEA = 0.059, SRMR = 0.081, CFI = 0.973 and TLI = 0.967). However, the SRMR was slightly above the cutoff value of 0.08. Therefore, to improve the model fit of Model 1.1, we examined modification indices (Supplementary Table S1) and respecified the model. We ran another first-order CFA (Model 1.2; Table 4), the respecified Model 1.1, by adding a covariance path between the pair of disturbance (*D*) terms: $D_{HSCL-11-SW*}$, $D_{HSCL-12-SW*}$, $D_{HSCL-13-SW*}$ and $D_{HSCL-10*}$ and $D_{HSCL-6*}$ and $D_{HSCL-7*}$ (Figure 2), which improved the chi-square of the Model 1.2 by ~53%. Adding these covariance

paths between disturbance terms substantially improved the model fit (RMSEA = 0.040, SRMR = 0.067, CFI = 0.989 and TLI = 0.985) (Table 4).

### Convergent validity

#### Fear Anxiety factor
The standardized factor loading point estimates ranged from 0.589 HSCL-1* ("Suddenly scared for no reason") to 0.870 HSCL-5* ("Heart pounding or racing") (Table 5 and Figure 2). All standardized factor loadings had a 90% ULCI of >0.5. The Fear Anxiety factor, on average, accounts for 55.5% (average variance extracted (AVE) = 0.56, 90% ULCI >0.5; Table 5) of the variance of the five latent response variables (Table 5). The omega coefficient subscale (ω1) was 0.763, which was above 0.7. All criteria for convergent validity were met for the Fear Anxiety factor.

#### Cognitive-Somatic Anxiety factor
The standardized factor loading point estimates ranged from 0.378 HSCL-7* ("Feeling tense") to 0.701 HSCL-3* ("Faintness, dizziness or weakness") (Table 5 and Figure 2). All standardized factor loadings had a 90% ULCI of >0.5, except for HSCL-7* ("Feeling tense"), which had a 90% ULCI value of 0.447. On average, the Cognitive-Somatic Anxiety factor explained 34.2% of the variance in the eight latent response variables (AVE = 0.342, 90% ULCI <0.5; Table 5). The omega coefficient subscale (ω₂) was 0.706, exceeding the recommended threshold of 0.7. Overall, almost all criteria for convergent validity of the Cognitive-Somatic Anxiety factor were met, except that the AVE had a 90% ULCI <0.5 and the standardized loading for HSCL-7* had a 90% ULCI <0.5.

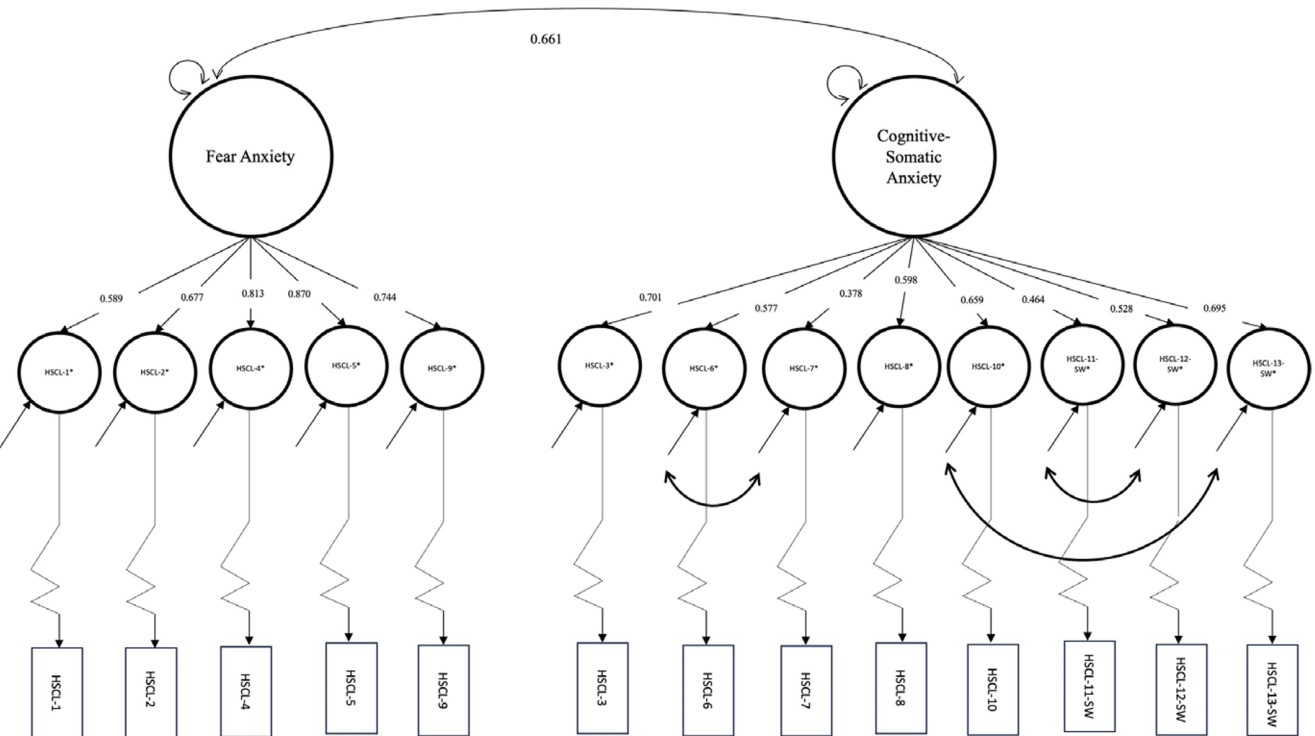

**Figure 2.** The two-factor (Fear Anxiety and Cognitive-Somatic Anxiety) HSCL-10-SW model (Model 1.2, *n* = 403) with estimated standardized factor loadings and disturbance term covariances using the DWLS with robust standard errors and mean-and-variance adjusted estimator.

| Latent response variables | Observed variables | Items |
|---|---|---|
| HSCL-1* | HSCL-1 | 1. Suddenly scared for no reason |
| HSCL-2* | HSCL-2 | 2. Feeling fearful |
| HSCL-4* | HSCL-4 | 4. Nervousness |
| HSCL-5* | HSCL-5 | 5. Heart pounding or racing |
| HSCL-9* | HSCL-9 | 9. Spells of terror |
| HSCL-3* | HSCL-3 | 3. Faintness, dizziness or weakness |
| HSCL-6* | HSCL-6 | 6. Trembling |
| HSCL-7* | HSCL-7 | 7. Feeling tense |
| HSCL-8* | HSCL-8 | 8. Headaches |
| HSCL-10* | HSCL-10 | 10. Feeling restless, cannot sit still |
| HSCL-11-SW* | HSCL-11-SW | 11. Irritability |
| HSCL-12-SW* | HSCL-12-SW | 12. Lost in thoughts/overthinking |
| HSCL-13-SW* | HSCL-13-SW | 13. Forgetfulness |

### Discriminant validity

Inter-item polychoric correlations between the HSCL-10-SW and PHQ-8 (Figure 3) ranged from 0.07 (e.g., HSCL-4* ("Suddenly scared") vs. PHQ-4* ("Fatigue"); HSCL-9* ("Spell of terror") vs. PHQ-5* ("Appetite change"); HSCL-9* ("Spell of terror") vs. PHQ-1* ("Anhedonia")) and 0.50 (e.g., HSCL-10* ("Feeling restless, can't sit still") vs. PHQ-7* ("Concentration difficulties")). The overall fit of Model 2 (HSCL-10-SW Model 1.2 and the one-factor PHQ-8 model) was acceptable (RMSEA = 0.058, SRMR = 0.078, CFI = 0.974, and TLI = 0.970). The correlation between Depression and Fear Anxiety constructs was ($\rho_1 = 0.571$, 90% LLCI<0.8, Supplementary Table S2); this can indicate that Depression and Fear Anxiety factors were empirically distinct. In contrast, the correlation between Depression and Cognitive-Somatic Anxiety factors ($\rho_2 = 0.923$, 90% LLCI > 0.9, Supplementary Table S2) suggests that these two pairs of constructs may not be empirically distinct. Therefore, discriminant validity between Depression and Fear Anxiety factors was conclusive.

**Table 4.** Overall model fit using DWLS with robust standard errors and mean-and-variance adjusted estimator

| Model | Chi-square | df | p | RMSEA | RMSEA 90% CI | SRMR | CFI | TLI |
|---|---|---|---|---|---|---|---|---|
| Model 1.1 (*n* = 403) | 153.623 | 64 | <0.001 | 0.059 | [0.047–0.071] | 0.081 | 0.973 | 0.967 |
| Model 1.2 (*n* = 403) | 99.412 | 61 | 0.001 | 0.040 | [0.025–0.053] | 0.067 | 0.989 | 0.985 |
| Model 2 (*N* = 809) | 607.773 | 179 | <0.001 | 0.054 | [0.050–0.059] | 0.074 | 0.977 | 0.973 |

*Note*: CFI, comparative fit Index; RMSEA, root mean square error approximation; SRMR, standardized root mean square residual; TLI, Tucker–Lewis Index; Model 1.1 was the initial HSCL-10-SW model. Model 1.2 was the respecified model with correlated disturbance terms. Model 2 was the HSCL-10-SW (Model 1.2) and the one-factor PHQ-8 model used for discriminant validity assessment, using all observations (*N* = 809).

**Table 5.** DWLS with robust standard errors and mean-and-variance adjusted parameter estimates of the HSCL-10-SW (*n* = 403, Model 1.2)

| Parameter | Unstandardized | | Standardized | | 90% CI | |
|---|---|---|---|---|---|---|
| | Estimate | SE | Estimate | SE | | |
| **Fear Anxiety factor** | | | | | | |
| Fear Anxiety → HSCL-1* ("Suddenly scared for no reason") | 1.000 | – | 0.589 | 0.034 | [0.533 | 0.644] |
| Fear Anxiety → HSCL-2* ("Feeling fearful") | 1.150 | 0.085 | 0.677 | 0.030 | [0.627 | 0.726] |
| Fear Anxiety → HSCL-4* ("Nervousness") | 1.381 | 0.093 | 0.813 | 0.028 | [0.767 | 0.858] |
| Fear Anxiety → HSCL-5* ("Heart pounding or racing") | 1.478 | 0.101 | 0.870 | 0.029 | [0.822 | 0.919] |
| Fear Anxiety → HSCL-9* ("Spells of terror") | 1.264 | 0.097 | 0.744 | 0.037 | [0.683 | 0.805] |
| $\omega_1$= 0.763 AVE = 0.555 90% CI [0.482–0.634] | | | | | | |
| **Cognitive-Somatic Anxiety factor** | | | | | | |
| Cognitive-Somatic Anxiety → HSCL-3* ("Faintness, dizziness or weakness") | 1.000 | – | 0.701 | 0.033 | [0.647 | 0.756] |
| Cognitive-Somatic Anxiety → HSCL-6* ("Trembling") | 0.823 | 0.068 | 0.577 | 0.037 | [0.517 | 0.638] |
| Cognitive-Somatic Anxiety → HSCL-7* "(Feeling tense") | 0.539 | 0.066 | 0.378 | 0.042 | [0.309 | 0.447] |
| Cognitive-Somatic Anxiety → HSCL-8* ("Headaches") | 0.852 | 0.063 | 0.598 | 0.031 | [0.546 | 0.649] |
| Cognitive-Somatic Anxiety → HSCL-10* ("Feeling restless, can't sit still") | 0.940 | 0.063 | 0.659 | 0.035 | [0.602 | 0.717] |
| Cognitive-Somatic Anxiety → HSCL-11-SW* ("Irritability") | 0.661 | 0.057 | 0.464 | 0.032 | [0.412 | 0.516] |
| Cognitive-Somatic Anxiety → HSCL-12-SW* ("Lost in thoughts/overthinking") | 0.752 | 0.060 | 0.528 | 0.032 | [0.475 | 0.580] |
| Cognitive-Somatic Anxiety → HSCL-13-SW* ("Forgetfulness") | 0.991 | 0.069 | 0.695 | 0.032 | [0.643 | 0.748] |
| $\omega_2$= 0.706 AVE = 0.342 90% CI [0.281–0.409] | | | | | | |
| **Covariance** | | | | | | |
| Fear Anxiety ⌣ Cognitive-Somatic Anxiety | 0.273 | 0.021 | 0.661 | 0.026 | [0.617 | 0.704] |
| **Disturbance term covariance** | | | | | | |
| $D_{HSCL-11-SW*}$ ⌣ $D_{HSCL-12-SW*}$ | 0.255 | 0.048 | 0.338 | 0.060 | [0.240 | 0.437] |
| $D_{HSCL-3*}$ ⌣ $D_{HSCL-10*}$ | −0.265 | 0.068 | −0.495 | 0.146 | [−0.732 | −0.254] |
| $D_{HSCL-6*}$ ⌣ $D_{HSCL-7*}$ | 0.293 | 0.079 | 0.388 | 0.102 | [0.221 | 0.555] |

*Note*: D = Disturbance term. The curved bidirectional arrows indicate covariances. $D_{HSCL-11-SW*}$ = disturbance term of the latent response variable "Irritability," $D_{HSCL-12-SW*}$ = disturbance term of the latent response variable "Lost in thoughts/overthinking," $D_{HSCL-3*}$ = disturbance term of the latent response variable "Faintness, dizziness or weakness," $D_{HSCL-10*}$ = disturbance term of the latent response variable "Feeling restless, can't sit still," $D_{HSCL-6*}$ = disturbance term of the latent response variable "Trembling," and $D_{HSCL-7*}$ = disturbance term of the latent response variable "Feeling tense."

## Discussion

The primary objective of this study was to assess the psychometric properties of the HSCL-10-SW in the Bay of Ranobe, in southwestern Madagascar. We found a two-factor solution (Fear Anxiety and Cognitive-Somatic Anxiety) that characterized anxiety disorders in the HIARA cohort adult participants in October 2023. The validation of an anxiety scale in southwestern Madagascar provides a path forward to diagnosing anxiety disorders in general populations in Madagascar by nonclinicians.

### Factor structure

The two-factor structure derived from the EFA, Fear Anxiety and Cognitive-Somatic Anxiety factors, was consistent with findings from other studies assessing the factor structure of the HSCL-25 (Kaaya et al., 2002; Al-Turkait et al., 2011), which were initially proposed by Clark and Watson (1991) based on the tripartite model. Broadly, one factor predominantly consisted of symptoms of fearfulness (e.g., "Suddenly scared for no reason," "Feeling fearful," and "Spell of terror"), whereas the other factor consisted of hyperarousal symptoms (e.g., "Faintness, dizziness or weakness," "Trembling," "Heart pounding or racing," and "Feeling tense"). In contrast to previous findings, the item "Heart pounding or racing" loaded on the Fear Anxiety factor rather than the Cognitive-Somatic Anxiety factor. This may be attributed to the cultural conceptualizations of the mind and heart in some Sub-Saharan African countries, where the two are regarded as tightly connected, and the heart is viewed as the source of emotions such as fear and anger (Patel, 1995). In addition, the "Heart pounding or racing" item was strongly correlated with the Fear Anxiety factor. This suggests that this item was the best indicator of anxiety disorders for our participants in the Bay of Ranobe, in southwestern Madagascar.

We added three items (*irritability*, *lost in thoughts/overthinking* and *forgetfulness*) derived from a previous study (Randriamady et al., 2025) to supplement the original HSCL-10 anxiety subscale. The three items all loaded onto the Cognitive-Somatic Anxiety factor. Among the three items, *forgetfulness* was strongly correlated with the Cognitive-Somatic Anxiety factor, followed by *lost in thoughts/ overthinking*, which was the second most frequently endorsed item on the HSCL-10-SW.

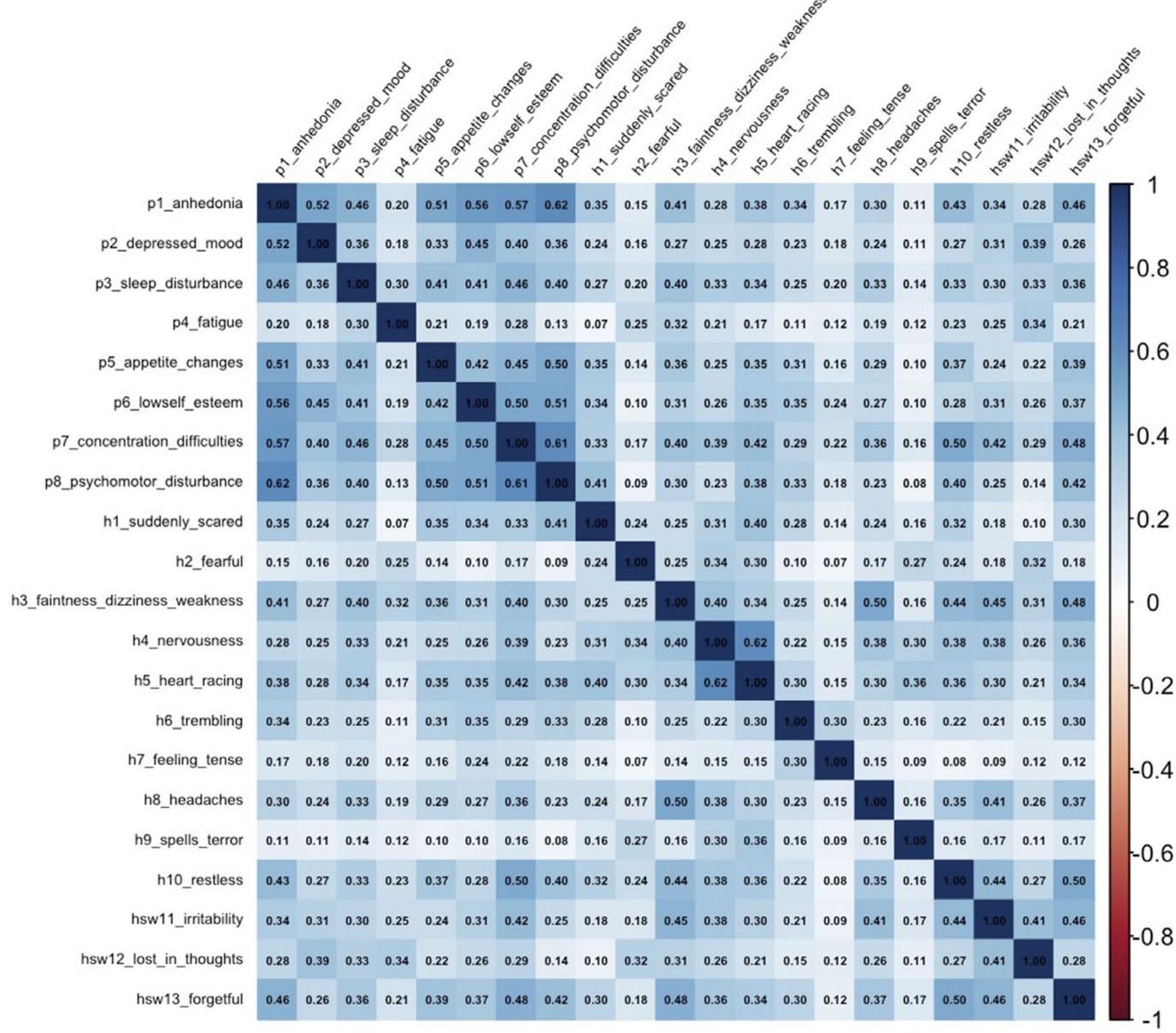

**Figure 3.** The heatmap polychoric correlation between HSCL-10-SW and PHQ-8 latent response variables (*N* = 809).

Although *irritability* was the second most endorsed item, its correlation with the Cognitive-Somatic Anxiety factor was moderate. This result was not consistent with Kayaa et al.'s (2008) study in Tanzania, in which *irritability*, along with social withdrawal, accounted for the largest share of variance in the Dar-es Salaam Symptom Questionnaire (a culturally adapted version of the HSCL-25). Several factors may explain this discrepancy. First, this contradiction might be attributed to the methods used. We used CFA instead of EFA. Thus, the results differed. Second, we administered the HSCL-10-SW to the general population, rather than an antenatal population (i.e., all women). Third, the composition of the adapted HSCL items differed significantly, as we focused only on the HSCL-10 anxiety subscale.

*Irritability* is also a marker of generalized anxiety disorder (American Psychiatric Association, 2022), and is found in analogous scales measuring anxiety disorders, such as the GAD-7 and the Mood and Anxiety Symptom Questionnaire (MASQ) (Watson et al., 1995a, 1995b; Spitzer et al., 2006). Similarly, *lost in*

*thoughts/overthinking* could be anxiety-specific symptoms. That idiom of distress can be interpreted as excessive anxiety and worry, which are part of the GAD symptom criteria (American Psychiatric Association, 2022). Specifically, the *overthinking* is similar to the "Thinking too much" idiom, which can be broadly interpreted as rumination, worry or intrusive thoughts, and is often associated with *forgetfulness* (Kaiser et al., 2015). Therefore, supplementing the HSCL-10 anxiety subscale with these items may improve the accuracy of anxiety disorders screening in the Bay of Ranobe, southwestern Madagascar. More broadly, this underscores the importance of including culturally relevant symptoms associated with anxiety disorders, which are often absent from standard scales.

### Validity

We assessed the psychometric validity of the culturally adapted HSCL-10-SW and found that it demonstrated acceptable convergent validity. Almost all standardized factor loadings for both the

Fear Anxiety and Cognitive-Somatic Anxiety factors had a 90% ULCI>0.5, except for the "Feeling tense" item that loaded onto the cognitive-somatic factor. Our findings are consistent with Lee et al. (2008), who also found low cross-loadings associated with the "Feeling tense" item when assessing the factor structure of the original HSCL-10 anxiety subscale, and using an EFA, not a CFA. A possible explanation for this low standardized factor loading may be attributed to how the "Feeling tense" item was translated in the Vezo and Masikoro dialects. Unlike the other items, the "Feeling tense" item underwent an iterative translation process; however, its final wording may still be misunderstood by participants. An alternative explanation, besides translation challenges, was that *feeling tense* might be culturally interpreted as physical tension. The association between Cognitive-Somatic Anxiety and "Feeling tense" may therefore be confounded by factors such as physical labor (e.g., fishing or farming) or illness, which could lead to an underestimation of the correlation. It is also possible that participants were less likely to report somatic symptoms compared to cognitive symptoms. For instance, a study found that Chinese participants from Changsha and Western (Euro-Canadian) participants from Toronto significantly reported somatic and psychological symptoms differently. Chinese participants reported more somatic symptoms than the Euro-Canadian participants. However, somatic symptoms were interpreted similarly (Ryder et al., 2008).

We also found that the AVE of the Cognitive-Somatic Anxiety factor was below the cutoff (90% ULCI <0.5) suggested by Cheung et al. (2023). However, a lower AVE is usually permissible in general, and an AVE ~ 0.40 is considered "very good" (Kline, 2023). Therefore, although we did not fully meet Cheung et al.'s (2023) stringent criteria, we still had acceptable convergent validity for the HSCL-10-SW.

We also evaluated the discriminant validity of the HSCL-10-SW and whether anxiety disorders were empirically distinct from depression (PHQ-8). Our discriminant validity assessment revealed that Depression and Fear Anxiety factors of the HSCL-10-SW were empirically distinct. This was demonstrated with the low correlation between Depression and Fear Anxiety factors. In contrast, we found poor discriminant validity between Depression and the Cognitive-Somatic Anxiety factors. Indeed, our results did not support Watson et al. (1995b) findings, as they found that a factor they labeled the *anxious arousal* factor, similar to the Cognitive-Somatic Anxiety factor, should be distinct from the Depression factor. They contended that hyperarousal symptoms were unique to anxiety disorders. Therefore, they should not overlap with other depressive symptoms, such as anhedonia. This discrepancy may also be attributable to differences in the measures used to assess discriminant validity, as well as to the constellation of symptoms represented in those measures. In the present study, we compared the PHQ-8 and the HSCL-10-SW, whereas Watson et al. (1995b) administered a single instrument, the MASQ, which assesses depression, anxiety and general distress symptoms within one scale. This inconsistency could be due to the method used to assess discriminant validity. Indeed, our analyses examined correlations between underlying latent constructs using a CFA framework, rather than correlations between observed subscale scores. These approaches are conceptually distinct and may yield different estimates of discriminant validity.

Finally, a possible explanation for the poor discriminant validity between Depression and Cognitive-Somatic Anxiety factors might be that the two factors stem from the same underlying construct. It is probable to hypothesize that the underlying construct itself is

major depressive disorder with anxious distress (American Psychiatric Association, 2022). The reasons are that the "Feeling tense" and "Feeling restless, can't sit still" items are specifiers of major depression with anxiety distress, but in addition to that, the "Irritability" is also an associated feature of major depressive disorder (American Psychiatric Association, 2022).

### Limitations and future research

More qualitative research should be undertaken to identify anxiety-like syndromes in the Bay of Ranobe, in southwestern Madagascar, that encompass anxiety disorder symptoms. Unlike the depressive-like syndromes, anxiety-like syndromes did not emerge naturally from a previous mixed-method study that elicited mental health syndromes in southwestern Madagascar (Randriamady et al., 2025). The three symptoms we used to supplement the original HSCL-10 were derived from depressive-like, grief-like and general distress-like syndromes (Randriamady et al., 2025). Additionally, the majority of the HSCL-10-SW items, mostly the HSCL-10 anxiety subscale, were translated from English into Vezo and Masikoro dialects. In addition, the HSCL-10-SW should not be compared with the original HSCL-10 anxiety subscale, as the three additional items were unique to the Bay of Ranobe in southwestern Madagascar. Therefore, the HSCL-10-SW is a context-specific scale for southwestern Madagascar, which may be potentially generalizable to Madagascar but not as a modified version of the HSCL-10 anxiety subscale.

A few aspects of validity were not assessed in this study. First, we did not conduct a criterion validity assessment to establish an appropriate cutoff score for identifying probable anxiety disorders using the HSCL-10-SW. Therefore, the HSCL-10-SW should be used solely as a screening scale for anxiety disorder symptoms rather than for diagnosing probable anxiety disorders. Second, we did not assess predictive validity by gauging, for instance, the association between HSCL-10-SW scores and food insecurity levels. That assessment would have been particularly relevant given the growing body of evidence linking food insecurity to mental health symptoms and the prevalence of food insecurity in that region (Golden et al., 2024b; Kelahan et al., 2026). Despite these limitations, the HSCL-10-SW is a reliable and valid scale for measuring anxiety disorder symptoms in the Bay of Ranobe, southwestern Madagascar.

Finally, we have not assessed measurement invariance across groups in this study. For instance, Vezo, Masikoro and Antandroy might have different concepts of anxiety disorders. Thus, we cannot meaningfully compare the scores between these groups. While we have not conducted measurement invariance analysis for the HSCL-10-SW, we found in our previous study that the concept of depression measured by the PHQ-8 was the same across different groups (Randriamady et al., 2025). Therefore, we might anticipate that it would be the same for the anxiety disorders measured by the HSCL-10-SW.

### Conclusions

In Madagascar, widespread poverty and economic instability limit access to mental health services, nutritious food and stable housing, all of which are essential for psychological well-being. Political unrest and weak governance contribute to chronic stress and uncertainty, particularly among youth and marginalized communities. Additionally, climate-related disasters such as cyclones and

droughts displace families and destroy livelihoods, exacerbating trauma, anxiety and depression across vulnerable populations. The HSCL-10-SW is a reliable and valid measure to screen for anxiety disorder symptoms in the Bay of Ranobe, in southwestern Madagascar, and could potentially serve as a model for assessing anxiety disorders in other regions of the country. However, we caution the general, scaled use of the HSCL-10-SW before further cultural validation and qualitative studies can definitively determine the utility of this diagnostic tool.

**Open peer review.** To view the open peer review materials for this article, please visit http://doi.org/10.1017/gmh.2026.10185.

**Supplementary material.** The supplementary material for this article can be found at http://doi.org/10.1017/gmh.2026.10185.

**Data availability statement.** Data requests should be addressed to the first author at hrandriamady@g.harvard.edu.

**Acknowledgments.** The authors would like to thank the Institut Halieutique et des Sciences Marines (IHSM), Professor Gildas Todinanahary, Emma Gibbons and Reef Doctor for their logistical support. The authors would also like to thank Dr. Kathy Trang at the Department of Epidemiology, Harvard TH Chan School of Public Health, for her assistance in interpreting the qualitative data. The authors would also like to thank Celina Razanajaosoa for her help in back-translating the HSCL-10-SW scale. The authors are grateful to Dr. Fabien Rakotondramanana from the Ministry of Public Health in Toliara. The authors would like to thank Dr. Vola Nirina Andrianavalona and Dr. Nivohanitra Razafindrasoa from the Ministry of Public Health in Antananarivo for conducting the initial focus group discussions. Above all, the authors would like to thank all participants in the study and the communities of the Bay of Ranobe.

**Author contribution.** **HJR:** Research conception and design, collection of data, analysis of data, interpretation of data, writing the manuscript, review and editing the manuscript; **MS:** Interpretation of data, writing the manuscript, review and editing the manuscript; **RES:** Research conception and design, and review and editing the manuscript; **AFM:** Research conception and design, collection of data, review and editing the manuscript; **R:** Collection of data, review and editing the manuscript; **MR:** Collection of data, review and editing the manuscript; **NVV:** Collection of data, review and editing the manuscript; **FD:** Collection of data, review and editing the manuscript; **MYS:** Collection of data, review and editing the manuscript; **JCM:** Collection of data, review and editing the manuscript; **HOR:** Collection of data, review and editing the manuscript; **KCK:** Research conception and design, writing the manuscript, review and editing the manuscript; **CDG:** Research conception and design, collection of data, writing the manuscript, review and editing the manuscript.

**Financial support.** Financial support for this study was provided by Belmont Forum through the National Science Foundation (RISE-2022717 CDG), the Harvard University Center of African Studies Graduate Travel Grants (HJR), Rose Service Learning Fellowship (HJR), and the Harvard President's Climate Change Solutions Fund (CDG and KCK).

**Competing interests.** The authors declare no competing interests.

**Use of AI.** The authors used ChatGPT (OpenAI) to assist with English grammar checking and language editing. The tool was used solely to improve clarity and readability.

**Ethical statements.** All participants were recruited and enrolled following our IRB-approved study (Protocol #20–1944 and 22–0491, Committee on the Use of Human Subjects, Office of Human Research Administration at the Harvard T.H. Chan School of Public Health). The study was also reviewed and approved by the Ethics Committee of the Ministry of Public Health (N036MSANP/SG/AMM/CERBM), and then reviewed and stamped by the Division of Mental Health Services at the Malagasy Ministry of Health and by the local medical inspector in Toliara II.

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
