## [Reviewer Report]

I have read the referred article with keen interest. The information is interesting and innovative; conclusion section is interesting and authors can improve it further. I am recommending authors to do a little more work and add latest literate to support the study. The authors need to improve results section. The level of English is good and smooth, e.g., the language standard, specifically the grammar, of sufficient quality to meet scientific merit for publication. However, I suggest authors to double check for language quality. Describe scientific contribution of the study to the existing body of knowledge. I endorse this manuscript after minor revision as suggested. The topic is interesting and worthy of attention. The methodology is adequate and the conclusions are consistent with the reported data. The manuscript can be improved by expanding the references and citing some recently published articles on this topic.

I also suggest that the authors highlight the scientific contribution of their work to the existing body of knowledge and expand the reference list to include some recent studies relevant to this topic, such as:

Sarfraz, R., Aqeel, M., Lactao, D. J., & Khan, D. S. (2021). Coping Strategies, Pain Severity, Pain Anxiety, Depression, Positive and Negative Affect in Osteoarthritis Patients; A Mediating and Moderating Model. Nature-Nurture Journal of Psychology, 1(1 SE-), 18–28. https://thenaturenurture.org/index.php/psychology/article/view/8

The Association of Maladaptive Coping Strategies with Adverse Parenting Styles and Symptoms of Mood Swings, Stress, Anxiety, and Depression in Patients with Conversion Disorder …

A hindrance to proper health care: psychometric development and validation of opiophobia questionnaire among doctors in Pakistan

Immediate Calamity Based Distress: Psychometric Development and Validation of Fear of Affliction Scale

---

## [Reviewer Report]

Thanks for the opportunity to review this interesting manuscript on validation of an adapted HSCL-10 in a sample of adults in Madagascar. Overall, this is a strong study which makes a significant contribution to the very sparse literature on anxiety in Madagascar. The analysis is robust and the manuscript is largely clearly written.

It would be useful to provide some more information on the translation process, why the two dialects were chosen, and how the differences between the dialects influenced the implementation of the measure, and what proportion of participants were administered the measure in each dialect. It would also be useful to explicate if there should have been translation to Antandroy given the large proportion of participants of this ethnicity, or if this is covered by the translation to Vezo and Masikoro dialects. Similarly, why not: “Additionally, the majority of the HSCL-10-SW items were translated from English into Vezo and Masikoro dialects, but not into the terminology or vernacular of the Bay of Ranobe.”

There is only one measure being used for discriminant validity (PHQ-8). There’s no convergent or predictive validity. It would be useful to add an extended discussion of this to the limitations.

Given the very limited literature on mental health in Madagascar in general, it would be useful to more clearly explain in the Discussion what the findings of this study add to understanding of and prevalence of mental health in Madagascar generally.

Small points:

- It would be useful in the abstract to provide some more demographic details on the participants: percent by gender, mean age, indication of the area of Madagascar where the data was collected (more specifically than the “southwestern” of the title).

- Under Study Participants, add more information on where these coastal and inland communities are. That is: what region? A slightly deeper description of the context of the participants would be helpful.

- I would recommend moving Table 1 up to the Study Participants section and adding gender to it. Please also add mean and standard deviation of age in section 3.1.

- “starled” should be “startled” I believe.

- “only “spells of terror” were retained.” should change to “was”

- This repetition is not necessary: “using Cheung et al. 286 (2023) and Rönkkö and Cho (2022) (Cheung et al. 2023; Rönkkö and Cho 2022).”

---

## [Reviewer Report]

Comments

Methods

1. Justification for Item Expansion

o While the rationale for adding the three culturally derived items is strong, it would be helpful to explicitly discuss how adding these items affects comparability with the original HSCL-10. Consider clarifying whether the HSCL-10-SW is intended as a context-specific instrument or as a modified version suitable for broader use in similar settings.

2. Details on Psychometric Analyses

o The methodology mentions EFA and CFA, but this section would benefit from additional detail, such as:

• Criteria for factor retention (e.g., eigenvalues, scree plot, parallel analysis).

• Estimation methods and rotation used in EFA.

• Model fit indices planned for CFA and thresholds for acceptable fit.

o Clarify whether EFA and CFA were conducted on separate samples or the same dataset, and justify this choice.

3. Validity Assessment

o The description of convergent and divergent validity using the PHQ-8 is appropriate, but it would be useful to:

• Explicitly state hypotheses (e.g., expected magnitude and direction of correlations).

• Clarify whether divergent validity was assessed using PHQ-8 subcomponents or total score only.

4. Ethical Considerations

o The manuscript notes verbal consent and IRB approval; consider briefly explaining why verbal rather than written consent was used, especially given international ethical review standards.

5. Naming Inconsistency (HSCL-10-SW vs. 13 Items)

There is a conceptual inconsistency in referring to the instrument as the “10-item Hopkins Symptom Checklist” while describing it as a 13-item scale. Consider renaming the instrument (e.g., HSCL-13-SW), or clearly state that HSCL-10-SW refers to the core anxiety subscale, with three additional culturally derived items analyzed alongside it. As written, this may confuse readers and reviewers regarding scale length and scoring.

6. Clarify Scoring and Use in Analyses

Specify whether all 13 items are summed into a single score, or whether the original 10 items and the three additional items are analyzed as separate factors or subscales. If different scoring approaches were used (e.g., total score vs. factor scores), this should be explicitly stated.

Data analysis

Factor Analysis Terminology

1. You refer to DWLS estimation for EFA, but DWLS is more commonly associated with CFA. Clarify whether:

o EFA was conducted using a polychoric correlation matrix with a factor extraction method appropriate for ordinal data (e.g., weighted least squares or minimum residuals), or

o EFA was implemented within a SEM framework.

Factor Retention Criteria

2. The scree plot is mentioned, but parallel analysis a commonly recommended method was not discussed. If parallel analysis was not conducted, briefly justify why the scree plot was deemed sufficient. Also clarify whether theoretical interpretability influenced factor retention decisions.

Bartlett’s Test Specification

3. Bartlett’s test is described as being conducted on Pearson’s correlation matrix, while the analysis treats data as ordinal. Consider clarifying whether Bartlett’s test was applied to a polychoric correlation matrix, or justify the use of Pearson correlations here.

Discussion

1. Several parts of the Discussion include detailed statistical information (e.g., exact correlations, confidence intervals, and item endorsement rankings). These details are more appropriate for the Results section. The Discussion would be strengthened by focusing on interpretation, theoretical integration, and implications rather than reporting numerical findings.

2. The discussion of the three added items is valuable, but their role could be framed more clearly in terms of what they reveal about anxiety expression in this setting rather than their statistical performance alone. Consider discussing whether these items reflect anxiety-specific symptoms or broader distress constructs, and how this may affect scale interpretation.

3. The explanation linking the lower loading of the “Feeling tense” item to translation challenges is plausible and important. However, it may be useful to acknowledge alternative explanations (e.g., cultural salience of bodily tension, overlap with physical labor or illness) to avoid attributing the issue solely to translation.

4. The discussion of divergence from Clark and Watson’s (1995) tripartite model is valuable. Consider more explicitly discussing whether this divergence reflects cultural variation in symptom structure, measurement differences, or the inclusion of culturally adapted items.